

# Evaluation of the facial profile of skeletal Class III patients undergoing camouflage orthodontic treatment: a retrospective study

Xiaobei Li[1], Yuan Tian[2], Runzhi Guo[3], Weiran Li[3], Danqing He[3] and Yannan Sun[3]

[1] First Clinic Division, Peking University School and Hospital of Stomatology, Beijing, China
[2] Department of Operational and Development Office, Peking University School and Hospital of Stomatology, Beijing, China
[3] Department of Orthodontics, Peking University School and Hospital of Stomatology &National Center for Stomatology & National Clinical Research Center for Oral Diseases, Beijing, China

Corresponding authors
Danqing He,
hedanqing@bjmu.edu.cn
Yannan Sun, ynsun2012@126.com

## ABSTRACT

**Objective**. To identify objective metrics for evaluating the esthetics of facial profiles in skeletal Class III patients undergoing camouflage orthodontic treatment.

**Methods**. Eighty Asian–Chinese patients classified as skeletal Class III were included. Thirty cephalometric measurements of pre- and posttreatment cephalograms were analyzed. Ten orthodontists assigned visual analog scale (VAS) scores to the pre- and posttreatment profiles based on standardized lateral photographs. Correlations between subjective VAS scores and objective measurements were assessed using Pearson correlation and stepwise multiple linear regression analysis.

**Results**. Lower incisor (L1) protrusion, nasolabial angle, lower lip-E line distance, SNB angle, and L1 to AP plane were significantly correlated with VAS scores of pretreatment profiles of skeletal Class III patients. Factors such as retraction of the lower incisor, increased interincisal angle and overjet, reduction of lower lip-E line distance, as well as augmentation of the *Z* angle and nasolabial angle were significantly correlated with the changes in VAS scores post-camouflage orthodontic treatment. Stepwise multiple linear regression analysis revealed that pretreatment nasolabial angle, changes in the lower lip-E line distance, and pretreatment Pog-NB distance were the key factors influencing the posttreatment facial profile esthetics of skeletal Class III patients with camouflage orthodontic treatment.

**Conclusion**. Several cephalometric measurements correlate with subjective facial esthetic evaluations of skeletal Class III patients. Changes in lower lip prominence, the pretreatment nasolabial angle, and Pog-NB distance are the main factors related to facial esthetics in skeletal Class III patients after camouflage orthodontic treatment.

## INTRODUCTION

Skeletal Class III malocclusion arises due to mandibular prognathism, maxillary deficiency, or a combination, which negatively affects the oral function and facial esthetics of patients (*Perillo et al., 2016*). Enhancing the facial profile is a primary concern for most skeletal

Class III patients. Typically, these patients are advised to undergo a combination of orthodontic treatment and orthognathic surgery to rectify the skeletal deformity (*Ngan & Moon, 2015*). However, many patients are hesitant to opt for orthognathic surgery due to cost and the invasive nature of the procedure (*Park, Emamy & Lee, 2019*; *Araujo & Squeff, 2021*). As a result, some patients prefer camouflage orthodontic treatments to improve their facial profiles and establish functional occlusion. While successful cases have been documented (*Lin & Gu, 2003*; *Yang, Ding & Feng, 2011*; *Choi et al., 2022*; *He et al., 2022*), few studies have focused on identifying key objective indicators in facial profile evaluations of skeletal Class III patients, which will be crucial for informed clinical decision-making. Identifying these indicators will also be important for patients and orthodontists in terms of evaluating facial esthetics post-camouflage orthodontic treatment (*Burns et al., 2010*).

Numerous cephalometric analyses of hard and soft tissues have been devised to quantify facial esthetics. These include the esthetic plane (*Ricketts, 1968*), B line (*Hsu, 1993*), H-angle (*Holdaway, 1983*), $Z$-angle (*Merrifield, 1966*), and nasolabial angle (*Burstone, 1967*). Craniofacial morphology can also affect facial esthetics assessments (*Kasai, 1998*). For skeletal Class I patients, the maxillary incisor position plays a pivotal role in the evaluation of the facial profile (*El Asmar et al., 2020*; *He, Gu & Sun, 2020*). However, the critical hard and soft tissue measurements influencing the facial evaluation of skeletal Class III patients remain unclear. Whether the positions of the upper and lower incisors affect facial evaluations of skeletal Class III patients undergoing camouflage treatment also requires further exploration.

While judgements of facial attractiveness are inherently subjective, the use of a visual analog scale (VAS) for facial esthetics assessments based on photographs has proven both valid and reliable (*Kiekens et al., 2005*; *Kiekens et al., 2008*; *Shoukat Ali et al., 2021*). Some studies have compared VAS scores with objective indicators to evaluate the profile of skeletal Class I and II patients (*Huang & Li, 2015*; *Guo et al., 2023*). However, more work is still needed to explore correlations between subjective and objective evaluation of the post-treatment profile of skeletal Class III patients.

In sum, this study sought to determine key objective measurements that influence the improvement of facial profile esthetics of skeletal Class III patients who underwent camouflage orthodontic treatments, thereby providing a basis for clinical decision-making in the non-surgical treatment of skeletal Class III patients.

## MATERIALS AND METHODS

### Patients

Power analyses were performed using the mean and standard deviation (SD) of incisor mandibular plane angle (IMPA) change (mean 6.89° and SD 15°) from *Alhammadi et al. (2022)*. The minimum sample size was 40 to achieve 80% power for detecting a clinically relevant effect, using an $\alpha$-value of 0.05. We searched the archives for consecutive patients meeting the eligibility criteria. Ultimately, this retrospective study included 80 Asian-Chinese skeletal Class III patients who underwent camouflage orthodontic treatment from 2015 to 2021 at the Department of Orthodontics, Peking University School of Stomatology.
The Peking University School and Hospital of Stomatology Ethics Committee approved the study protocol (PKUSSIRB-202168141). The following selection criteria were used:

1. Pretreatment ANB angle $\leq 0°$;
2. Pre- and posttreatment cephalometric radiographs and standardized facial photographs of sufficient diagnostic quality;
3. Patients who declined the combined orthodontics and orthognathic surgery treatment and opted for camouflage orthodontics;
4. No craniofacial abnormalities, recognized syndromes, or history of orthognathic or cosmetic facial surgery;
5. All camouflage treatment was achieved with fixed appliances in both jaws without applying orthopedic force.

The study population included 55 females and 25 males (age range 13–35 (mean $18.74 \pm 6.21$) years). Informed consent was obtained from all participating adults or from the parents or legal guardians of minors.

Based on the pretreatment SNB angle that indicated the relative mandibular position with the skull, we categorized patients into non-mandibular protrusion (SNB angle $\leq 84°$, $n = 36$) and mandibular protrusion groups (SNB angle $>84°$, $n = 44$). Differences between these two groups with distinct skeletal patterns were analyzed. The vertical dimension may affect the facial profiles of skeletal Class III patients, so we also compared the differences between high-angle patients (MP/SN angle $>37°$, $n = 29$) and non-high-angle patients (MP/SN angle $\leq 37°$, $n = 51$). Considering the age range of the subjects in this study, we also compared differences between adolescents (initial age $<18$ years, $n = 24$) and adults (initial age $\geq 18$ years, $n = 56$).

## Subjective evaluation of facial esthetics

We obtained pre- and posttreatment lateral facial photographs, taken in the natural head position (NHP) with the lips in a resting position, from patient records. Facial esthetics were evaluated by a panel of 10 orthodontic clinicians (five men and five women aged 33–49 years) selected through stratified random sampling from senior, medium, and junior faculties at the Department of Orthodontics at Peking University School of Stomatology. The panel assessed standardized digital facial photographs under uniform conditions in a laboratory. The facial photographs were shown on slides with a VAS ranging from 0 (very unpleasing) to 100 (very pleasing), as described previously (*Huang & Li, 2015*). Pre- and posttreatment facial photographs of the same subject were not placed on the same slide to avoid possible bias. Photographs were randomized before the assessment. The final score was the average of 10 scores.

## Objective measurements of cephalometric radiographs

The primary investigator digitized and traced all pretreatment (T1) and posttreatment (T2) cephalograms using Cephalometric Tracing and Analysis software (Dolphin Imaging, Canoga Park, CA, USA). Magnification differences between cephalostats were corrected before data analysis using a ruler. All cephalometric radiographs were acquired using the same X-ray setup. FA-Fall and overjet were measured using Adobe Photoshop

(Adobe Systems, San Jose, CA, USA) as described previously (*He, Gu & Sun, 2020*). The head position in the cephalogram was reoriented 7° inferior to the SN plane with Sella registration. In total, 30 linear and angular measurements for soft and hard tissue were analyzed. The objective measurements are defined in Table S1.

### Reliability analysis

To assess the reproducibility and validity of subjective facial esthetics, three judges performed assessments in the same manner 1 month later. For cephalometric error testing, pre- and posttreatment cephalograms of 15 randomly selected patients were assessed by the same operator 1 month later. All intraclass correlation coefficients (ICC) for the repeated measurements were >0.80.

### Statistical methods

Statistical analysis was performed using IBM SPSS Statistics for Macintosh software (ver. 20.0; IBM, Armonk, NY, USA). The normality of distribution was analyzed (Table S2), and the mean and standard deviation (SD) of each variable was calculated for normally distributed measurements. Correlations between objective measurements and subjective VAS scores were evaluated by Pearson's correlation. The median and quartiles were calculated and the Wilcoxon signed-rank test was performed when the measurement distributions were not normally distributed. Levene's test for homogeneity of variance was performed when performing independent samples $t$-tests, $P > 0.05$. Stepwise multiple linear regression analysis was also conducted. $P < 0.05$ was considered significant.

## RESULTS

### Descriptive data for objective measurements and facial VAS scores

Descriptive data for the objective measurements from pre- and posttreatment cephalometric radiographs are presented in Table 1, along with facial esthetic VAS scores. Both means and SDs are listed, except for ANB and Wits, which are reported as medians and quartiles because the data distribution was skewed.

### Correlations between pretreatment objective measurements and pretreatment subjective VAS scores

Pearson correlations between pretreatment VAS scores and objective cephalometric measurements in descending order of absolute magnitude are presented in Table 2. Correlations between ANB, Wits, and subjective VAS scores were adjusted using Spearman correlation. Five measurements were significantly correlated with the VAS score of the facial profile. The nasolabial angle was positively correlated with VAS score ($r = 0.254$, $p = 0.023$), whereas L1-AP ($r = −0.331$, $p = 0.003$), lower lip to E-line ($r = −0.246$, $p = 0.028$), SNB ($r = −0.244$, $p = 0.029$), and L1/AP ($r = −0.233$, $p = 0.038$) were negatively correlated with VAS score.

Within the non-mandibular protrusion group, three lower incisor position measurements (L1/MP, L1/AP, and L1-AP) were negatively correlated with the VAS scores of the facial profile, with correlation coefficients of $−0.390$ ($p = 0.019$), $−0.358$ ($p = 0.032$), and $−0.330$ ($p = 0.049$), respectively. Additionally, the gonial jaw angle ($r = 0.381$, $p =$

**Table 1** Cephalometric characteristics in skeletal class III patients with camouflage orthodontic treatment.

| Variable | Pre treatment | | Post treatment | | Normal | | Sig |
|---|---|---|---|---|---|---|---|
| | Mean | SD | Mean | SD | Value | SD | |
| Z Angle | 75.81 | 7.66 | 78.53 | 6.65 | 80 | 9 | 0.000** |
| Lower Lip-E line | 1.26 | 2.64 | 0.08 | 2.42 | −2.0 | 2.0 | 0.000** |
| Upper Lip-E Line | −2.69 | 2.24 | −2.35 | 2.10 | 3.9 | 2.0 | 0.071 |
| Nose prominence | 18.47 | 2.09 | 18.58 | 2.08 | – | – | 0.529 |
| Nasolabial Angle | 93.84 | 9.08 | 92.96 | 9.87 | 102 | 8.0 | 0.280 |
| MentoLabial Angle | 145.12 | 10.67 | 140.57 | 12.65 | 120 | 1.0 | 0.000** |
| SNA | 81.11 | 3.34 | 81.90 | 3.28 | 82.8 | 4.0 | 0.000** |
| SNB | 82.92 | 3.62 | 83.19 | 3.55 | 80.1 | 3.9 | 0.133 |
| MP/SN | 35.47 | 5.46 | 35.41 | 5.44 | 32.5 | 5.2 | 0.578 |
| MP/FH | 27.98 | 5.44 | 27.84 | 5.62 | 31.1 | 5.6 | 0.347 |
| Gonial Jaw Angle | 128.20 | 8.27 | 128.39 | 8.19 | 133.4 | 6.7 | 0.410 |
| Y axis | 68.49 | 3.35 | 68.41 | 3.39 | 66.3 | 7.1 | 0.559 |
| LFH | 54.30 | 1.76 | 54.94 | 1.61 | 57.0 | – | 0.000** |
| Pog-NB | 0.30 | 1.70 | 0.72 | 1.61 | 1.0 | 1.5 | 0.001** |
| OP/SN | 16.65 | 5.04 | 14.30 | 5.14 | 16.1 | 5.0 | 0.000** |
| U1/SN | 110.58 | 7.28 | 116.05 | 7.19 | 105.7 | 6.3 | 0.000** |
| U1/AP | 24.89 | 6.94 | 30.29 | 5.73 | 28.0 | 4.0 | 0.000** |
| U1-AP | 5.33 | 2.52 | 6.84 | 1.97 | 6.0 | 2.3 | 0.000** |
| U1/NA | 29.48 | 6.35 | 34.15 | 6.35 | 22.8 | 5.7 | 0.000** |
| U1-NA | 6.72 | 2.17 | 7.95 | 1.85 | 6.0 | 4.4 | 0.000** |
| L1/MP | 82.14 | 7.39 | 76.67 | 9.31 | 92.6 | 7.0 | 0.000** |
| L1/AP | 23.32 | 5.17 | 17.85 | 6.51 | 22.0 | 4.0 | 0.000** |
| L1-AP | 5.77 | 2.55 | 3.54 | 2.14 | 2.7 | 1.7 | 0.000** |
| L1/NB | 20.54 | 6.77 | 15.27 | 7.33 | 30.3 | 5.8 | 0.000** |
| L1-NB | 4.57 | 2.38 | 2.80 | 2.34 | 6.7 | 2.1 | 0.000** |
| Interincisal Angle | 131.80 | 10.54 | 131.87 | 10.27 | 125.4 | 7.9 | 0.963 |
| FA-Fall | 4.81 | 4.64 | 5.81 | 4.87 | – | – | 0.000** |
| overjet | −0.27 | 2.87 | 3.40 | 1.40 | – | – | 0.000** |
| score | 64.19 | 9.89 | 73.15 | 6.73 | – | – | 0.000** |
| Variable | pre treatment | | post treatment | | Normal | | sig |
| | median | quartile | median | quartile | value | SD | |
| ANB# | −1.20 | 2.50 | −0.90 | 1.78 | 2.7 | 2.0 | 0.000** |
| Wits# | −7.60 | 4.90 | −5.55 | 3.95 | −1.0 | 1.0 | 0.000** |

**Notes.**
#ANB, and Wits were shown as a skewed distribution, the descriptive analysis of these data was adjusted to median and quartile instead of mean and SD. * $P < 0.05$, ** $P < 0.01$.

0.022) was positively correlated with VAS score. Conversely, in the mandibular protrusion group, only the distance from the lower lip to the E-line ($r = -0.276$, $p = 0.044$) was significantly correlated with the pretreatment facial profile VAS score, as shown in Table 3. However, the correlations between pretreatment VAS scores and objective cephalometric measurements did not differ significantly between high-angle and non-high-angle patients

**Table 2** Pearson correlation between visual analog scale (VAS) score of pretreatment profile and 30 cephalometric measurements.

| Variable | r | P | Order |
|---|---|---|---|
| L1-AP | −0.331 | 0.003** | 1 |
| Nasolabial Angle | 0.254 | 0.023* | 2 |
| LowerLip-E line | −0.246 | 0.028* | 3 |
| SNB | −0.244 | 0.029* | 4 |
| L1/AP | −0.233 | 0.038* | 5 |
| L1-NB | −0.218 | 0.052 | 6 |
| Gonial Jaw Angle | 0.195 | 0.083 | 7 |
| Pog-NB | 0.188 | 0.095 | 8 |
| ANB# | 0.186 | 0.099 | 9 |
| L1/NB | −0.171 | 0.130 | 10 |
| SNA | −0.162 | 0.152 | 11 |
| L1/MP | −0.141 | 0.212 | 12 |
| MP/SN | 0.140 | 0.215 | 13 |
| Wits# | −0.106 | 0.347 | 14 |
| MP/FH | 0.103 | 0.365 | 15 |
| FA-Fall | 0.101 | 0.372 | 16 |
| Y axis | 0.099 | 0.382 | 17 |
| Z angle | 0.092 | 0.419 | 18 |
| overjet | 0.087 | 0.445 | 19 |
| Interincisal ange | 0.083 | 0.463 | 20 |
| U1/SN | −0.083 | 0.466 | 21 |
| Nose prominence | −0.075 | 0.506 | 22 |
| MentoLabial Angle | −0.067 | 0.557 | 23 |
| OP/SN | 0.064 | 0.572 | 24 |
| U1/AP | 0.047 | 0.677 | 25 |
| U1-AP | 0.039 | 0.730 | 26 |
| LFH | −0.037 | −0.037 | 27 |
| UpperLip-E line | −0.014 | 0.902 | 28 |
| U1-NA | −0.011 | 0.922 | 29 |
| U1/NA | −0.008 | 0.943 | 30 |

**Notes.**
#ANB, and Wits were shown as a skewed distribution, the correlations between subjective VAS scores and objective measurements were assessed using Spearman correlation. * $P < 0.05$, ** $P < 0.01$.

(Table S3). No significant differences were observed between adolescent and adult patients (Table S4).

## Correlations between changes in objective measurements and changes in subjective VAS scores

Pearson correlations between changes in VAS scores (Δscore) and changes in objective measurements are presented in Table 4. Correlations between ΔANB, ΔWits, and subjective VAS scores were adjusted using Spearman correlation. Of 30 objective measurements, changes in 10 measurements were significantly correlated with the Δscore of the facial profile. Changes in the lower incisor inclination and protrusion (ΔL1-AP, $r = −0.439$, $p$

**Table 3** Pearson correlation between visual analog scale (VAS) score of pretreatment profile and 30 cephalometric measurements in non-mandibular protrusion patients and mandibular protrusion patients.

| Non-mandibular protrusion ($n = 36$) | | | | Mandibular protrusion ($n = 44$) | | | |
|---|---|---|---|---|---|---|---|
| **Variable** | **r** | **P** | **Order** | **Variable** | **r** | **P** | **Order** |
| L1/MP | −0.390 | 0.019[*] | 1 | LowerLip-E line | −0.276 | 0.044[*] | 1 |
| Gonial Jaw Angle | 0.381 | 0.022[*] | 2 | L1-AP | −0.268 | 0.079 | 2 |
| L1/AP | −0.358 | 0.032[*] | 3 | Z Angle | 0.224 | 0.143 | 3 |
| L1-AP | −0.330 | 0.049[*] | 4 | MentoLabial Angle | −0.189 | 0.219 | 4 |
| Nasolabial Angle | 0.318 | 0.059 | 5 | Wits[#] | −0.170 | 0.269 | 5 |
| L1-NB | −0.302 | 0.073 | 6 | L1-NB | −0.161 | 0.295 | 6 |
| L1/NB | −0.295 | 0.081 | 7 | Nasolabial Angle | 0.160 | 0.301 | 7 |
| LowerLip-E line | −0.252 | 0.138 | 8 | L1-AP | −0.142 | 0.358 | 8 |
| MP/SN | 0.227 | 0.183 | 9 | MP/SN | −0.140 | 0.363 | 9 |
| Interincisal angle | 0.214 | 0.210 | 10 | ANB[#] | 0.137 | 0.373 | 10 |
| MP/FH | 0.186 | 0.278 | 11 | Pog-NB | 0.135 | 0.382 | 11 |
| Z Angle | 0.166 | 0.333 | 12 | MP/FH | −0.124 | 0.423 | 12 |
| FA-Fall | 0.161 | 0.349 | 13 | OP/SN | −0.118 | 0.446 | 13 |
| Pog-NB | 0.136 | 0.429 | 14 | L1/NB | −0.096 | 0.535 | 14 |
| overjet | 0.130 | 0.450 | 15 | U1/AP | 0.092 | 0.554 | 15 |
| Nose prominence | −0.122 | 0.479 | 16 | Y axis | −0.077 | 0.619 | 16 |
| SNB | −0.114 | 0.509 | 17 | U1-NA | −0.070 | 0.653 | 17 |
| U1/SN | −0.091 | 0.598 | 18 | SNB | −0.067 | 0.665 | 18 |
| Wits[#] | 0.087 | 0.614 | 19 | LFH | −0.067 | 0.665 | 19 |
| ANB[#] | 0.085 | 0.622 | 20 | U1/SN | 0.052 | 0.737 | 20 |
| U1/NA | −0.083 | 0.631 | 21 | SNA | 0.046 | 0.767 | 21 |
| UpperLip-E line | −0.077 | 0.656 | 22 | overjet | 0.044 | 0.776 | 22 |
| Y axis | −0.061 | 0.722 | 23 | U1/NA | 0.042 | 0.786 | 23 |
| U1/AP | −0.059 | 0.733 | 24 | FA-Fall | 0.037 | 0.809 | 24 |
| U1-NA | −0.040 | 0.819 | 25 | UpperLip-E line | −0.035 | 0.824 | 25 |
| SNA | −0.031 | 0.857 | 26 | L1/MP | 0.025 | 0.873 | 26 |
| U1-AP | −0.030 | 0.863 | 27 | Nose prominence | −0.014 | 0.929 | 27 |
| MentoLabial Angle | 0.029 | 0.867 | 28 | Interincisal Angle | 0.010 | 0.946 | 28 |
| LFH | −0.014 | 0.934 | 29 | Gonial Jaw Angle | −0.005 | 0.972 | 29 |
| OP/SN | 0.004 | 0.982 | 30 | U1-AP | 0.001 | 0.993 | 30 |

**Notes.**

[#]ANB, and Wits were shown as a skewed distribution, the correlations between subjective VAS scores and objective measurements were assessed using Spearman correlation. * $P < 0.05$, ** $P < 0.01$.

$< 0.001$; $\Delta$L1-NB, $r = -0.417$, $p < 0.001$; $\Delta$L1/NB, $r = -0.360$, $p = 0.001$; $\Delta$L1/AP, $r = -0.357$, $p = 0.001$; $\Delta$L1/MP, $r = -0.344$, $p = 0.002$) were significantly associated with improved VAS scores of profile change. Similarly, a decrease in the distance from the lower lip to the esthetic plane ($\Delta$Lower lip-E line, $r = -0.384$, $p < 0.001$) and a change in the $Z$ angle ($\Delta Z$ angle, $r = 0.274$, $p = 0.014$) were significantly correlated with the VAS score of profile change. Increases in the interincisal angle ($\Delta$U1/L1, $r = 0.306$, $p = 0.006$) and overjet ($\Delta$overjet, $r = 0.231$, $p = 0.039$) were also significantly correlated with increased

**Table 4 Pearson correlation between changes in visual analog scale (VAS) score and changes in 30 cephalometric measurements.**

| Variable | r | P | Order |
|---|---|---|---|
| $\Delta$L1-AP | −0.439 | 0.000** | 1 |
| $\Delta$L1-NB | −0.417 | 0.000** | 2 |
| $\Delta$LowerLip-E line | −0.384 | 0.000** | 3 |
| $\Delta$L1/NB | −0.360 | 0.001** | 4 |
| $\Delta$L1/AP | −0.357 | 0.001** | 5 |
| $\Delta$L1/MP | −0.344 | 0.002** | 6 |
| $\Delta$ interincisal Angle | 0.306 | 0.006** | 7 |
| $\Delta$Z angle | 0.274 | 0.014* | 8 |
| $\Delta$ nasolabial angle | 0.236 | 0.035* | 9 |
| $\Delta$overjet | −0.231 | 0.039* | 10 |
| $\Delta$U1-AP | −0.202 | 0.073 | 11 |
| $\Delta$U1-NA | −0.180 | 0.111 | 12 |
| $\Delta$FA-Fall | −0.174 | 0.123 | 13 |
| $\Delta$U1/SN | −0.145 | 0.200 | 14 |
| $\Delta$U1/AP | −0.138 | 0.221 | 15 |
| $\Delta$UpperLip-E line | −0.131 | 0.248 | 16 |
| $\Delta$U1/NA | −0.124 | 0.271 | 17 |
| $\Delta$Pog-NB- | 0.124 | 0.275 | 18 |
| $\Delta$LFH | 0.119 | 0.293 | 19 |
| $\Delta$MP/SN | 0.111 | 0.111 | 20 |
| $\Delta$Y axis | 0.110 | 0.331 | 21 |
| $\Delta$SNA | −0.107 | 0.345 | 22 |
| $\Delta$SNB | −0.095 | 0.404 | 23 |
| $\Delta$Nose prominence | −0.087 | 0.443 | 24 |
| $\Delta$MP/FH | −0.083 | 0.463 | 25 |
| $\Delta$Wits[#] | −0.067 | 0.553 | 26 |
| $\Delta$OP/SN | 0.063 | 0.579 | 27 |
| $\Delta$MentoLabial Angle | −0.050 | 0.657 | 28 |
| $\Delta$Gonial Jaw Angle | −0.029 | 0.740 | 29 |
| $\Delta$ANB[#] | 0.001 | 0.992 | 30 |

**Notes.**
[#] $\Delta$ANB, and $\Delta$ Wits were shown as a skewed distribution, the correlations between subjective VAS scores and objective measurements were assessed using Spearman correlation. * $P < 0.05$, ** $P < 0.01$.

VAS scores. Interestingly, an increase in the nasolabial angle ($\Delta$nasolabial angle, $r = 0.236$, $p = 0.035$) was significantly associated with improved VAS scores of profile changes in skeletal class III patients undergoing camouflage treatment.

In the non-mandibular protrusion group, three measurements indicating the position of the upper incisor ($\Delta$FA-Fall, $r = -0.471$, $p = 0.004$; $\Delta$U1-AP, $r = -0.408$, $p = 0.013$; $\Delta$U1-NA, $r = -0.330$, $p = 0.049$) suggested that lingual movement of the upper incisor is beneficial for facial esthetics. Changes in lower incisor inclination and protrusion also significantly influenced VAS score changes ($\Delta$L1-NB, $r = -0.444$, $p = 0.007$; $\Delta$L1-AP, $r = -0.429$, $p = 0.009$; $\Delta$L1/NB, $r = -0.380$, $p = 0.022$; $\Delta$L1/MP, $r = -0.378$, $p =$

**Table 5** Pearson correlation between changes in visual analog scale (VAS) score and changes in 30 cephalometric measurements of non-mandibular protrusion patients and mandibular protrusion patients.

| Non-mandibular protrusion ($n = 36$) | | | | Mandibular protrusion ($n = 44$) | | | |
|---|---|---|---|---|---|---|---|
| Variable | r | P | Order | Variable | r | P | Order |
| ΔFA-Fall | −0.471 | 0.004** | 1 | ΔL1-AP | −0.430 | 0.004** | 1 |
| ΔL1-NB | −0.444 | 0.007** | 2 | ΔL1-NB | −0.370 | 0.013* | 2 |
| ΔLowerLip-E line | −0.442 | 0.007** | 3 | ΔL1/AP | −0.352 | 0.019* | 3 |
| ΔL1-AP | −0.429 | 0.009** | 4 | Δoverjet | 0.342 | 0.023* | 4 |
| ΔInterincisal Angle | 0.413 | 0.012* | 5 | ΔL1/NB | −0.324 | 0.032* | 5 |
| ΔU1-AP | −0.408 | 0.013* | 6 | ΔLowerLip-E line | −0.314 | 0.038* | 6 |
| ΔL1/NB | −0.380 | 0.022* | 7 | ΔL1/MP | −0.302 | 0.047* | 7 |
| ΔL1/MP | −0.378 | 0.023* | 8 | ΔZAngle | 0.289 | 0.057 | 8 |
| ΔL1/AP | −0.352 | 0.035* | 9 | ΔGonial Jaw Angle | −0.248 | 0.104 | 9 |
| ΔU1-NA | −0.330 | 0.049* | 10 | ΔWits# | −0.237 | 0.122 | 10 |
| ΔU1/AP | −0.314 | 0.062 | 11 | ΔNasolabial Angle | 0.225 | 0.143 | 11 |
| ΔZAngle | 0.300 | 0.075 | 12 | ΔInterincisal Angle | 0.211 | 0.170 | 12 |
| ΔUpperLip-E line | −0.285 | 0.093 | 13 | ΔLFH | 0.209 | 0.173 | 13 |
| ΔU1/SN | −0.271 | 0.110 | 14 | ΔMP/SN | 0.193 | 0.211 | 14 |
| ΔU1/NA | −0.253 | 0.137 | 15 | ΔSNB | −0.187 | 0.225 | 15 |
| ΔANB# | −0.223 | 0.192 | 16 | ΔMP/FH | −0.157 | 0.310 | 16 |
| ΔNasolabial Angle | 0.219 | 0.200 | 17 | ΔANB# | 0.128 | 0.407 | 17 |
| ΔGonial Jaw Angle | 0.192 | 0.261 | 18 | ΔNose prominence | −0.119 | 0.441 | 18 |
| ΔMentoLabial Angle | −0.177 | 0.302 | 19 | ΔY axis | 0.116 | 0.452 | 19 |
| ΔOP/SN | 0.160 | 0.352 | 20 | ΔSNA | −0.102 | 0.511 | 20 |
| Δoverjet | −0.146 | 0.394 | 21 | ΔPog-NB | 0.094 | 0.545 | 21 |
| ΔWits# | 0.128 | 0.456 | 22 | ΔU1-NA | −0.072 | .0.641 | 22 |
| ΔY axis | 0.104 | 0.544 | 23 | ΔMentoLabial Angle | 0.066 | 0.672 | 23 |
| ΔPog-NB | 0.103 | 0.552 | 24 | ΔU1/SN | −0.052 | 0.740 | 24 |
| ΔSNA | −0.092 | 0.592 | 25 | ΔU1-AP | −0.043 | 0.781 | 25 |
| ΔMP/FH | −0.083 | 0.632 | 26 | ΔU1/NA | −0.030 | 0.846 | 26 |
| ΔSNB | 0.035 | 0.839 | 27 | ΔUpperLip-E line | −0.024 | 0.878 | 27 |
| ΔMP/SN | 0.023 | 0.895 | 28 | ΔOP/SN | −0.018 | 0.908 | 28 |
| ΔNose prominence | −0.020 | 0.908 | 29 | ΔU1/AP | −0.007 | 0.966 | 29 |
| ΔLFH | 0.015 | 0.933 | 30 | ΔFA-Fall | 0.002 | 0.991 | 30 |

Notes.
# ΔANB, and ΔWits were shown as a skewed distribution, the correlations between subjective VAS scores and objective measurements were assessed using Spearman correlation.
* $P < 0.05$, ** $P < 0.01$.

0.023; ΔL1/AP, $r = -0.352$, $p = 0.035$). An increase in interincisal angle (ΔU1/L1 angle, $r = 0.413$, $p = 0.012$) and a decrease in distance from the lower lip to the esthetic plane (ΔLower lip-E line, $r = -0.384$, $p = 0.007$) were also significantly correlated with VAS score changes (Table 5).

However, in the mandibular protrusion group, only retraction of the lower incisor was significantly correlated with VAS score changes (ΔL1-AP, $r = -0.430$, $p = 0.004$; ΔL1-NB, $r = -0.370$, $p = 0.013$; ΔL1/AP, $r = -0.352$, $p = 0.019$; ΔL1/NB, $r = -0.324$, $p = 0.032$; ΔL1/MP, $r = -0.302$, $p = 0.047$). An increase in overjet (Δoverjet, $r = 0.342$, $p = 0.023$)

**Table 6  Stepwise multiple linear regression analysis for posttreatment profiles.**

| Independent variables | B | $\beta$ | t value | p value |
|---|---|---|---|---|
| Nasolabial Angle | .746 | .956 | 76.395 | .000 |
| ∆LowerLip-E line | −2.132 | −.065 | −5.206 | .000 |
| Pog-NB | 1.366 | .032 | 2.909 | .005 |

Notes.
Adjusted R2 = 0.991.
B, partial regression coefficient; $\beta$, standardised partial regression coefficient.

and a reduction in the distance from the lower lip to the esthetic plane (∆Lower lip-E line, $r = -0.314$, $p = 0.038$) were also significantly correlated with VAS score changes (Table 5). No significant differences were observed between adolescent and adult patients (Table S5).

## Stepwise multiple linear regression analysis for posttreatment profiles

Stepwise multiple linear regression analysis was conducted to identify the main factors affecting VAS scores of posttreatment facial profiles in skeletal Class III patients. The results revealed that pretreatment nasolabial angle, changes in the distance from the lower lip to the esthetic plane (∆Lower lip-E line), and pretreatment Pog-NB distance were the key factors affecting the VAS scores of posttreatment profiles (Table 6).

## DISCUSSION

Enhancing the facial profile of skeletal Class III patients following camouflage orthodontic treatment is a major consideration for both patients and orthodontists (Burns et al., 2010). However, the key factors influencing the facial evaluation of skeletal Class III patients remain undefined. This study assessed treatment outcomes of skeletal Class III patients undergoing camouflage orthodontic treatment using cephalometric radiographs (for objective assessment) and lateral photographs (for subjective assessment). We sought to establish correlations between objective and subjective evaluations of profile esthetics and to explore key objective measurements in facial esthetic evaluations of skeletal Class III patients.

Our investigation focused on the correlations between objective measurements and subjective evaluations of facial attractiveness in skeletal Class III patients before treatment. The key objective measurements correlating with facial profile esthetics were the lower incisor position, nasolabial angle, lower lip position, and SNB angle before treatment. We observed a significant negative correlation between the pre-treatment lower incisor angle and lateral facial profile esthetics. Previous studies have reported that the maxillary incisor position is a major factor contributing to facial profile evaluations of skeletal Class I and II patients (El Asmar et al., 2020; He, Gu & Sun, 2020; Guo et al., 2023). However, we found that for skeletal Class III patients, the underlying lower incisor position is more important for facial attractiveness. This might be due to a moderate compensatory protrusion of the maxillary incisor, which can enhance the facial profile of skeletal Class III patients, but excessive protrusion can diminish the nasolabial angle, thus reducing the

VAS score. Previous studies have reported that it is necessary to establish normal incisor inclination in patients with a protruded mandible (*Zarif Najafi et al., 2015*). Our results were consistent with these previous findings: the lower incisor position is a major factor contributing to facial profile evaluations of skeletal Class III patients. Interestingly, unlike other malocclusion types, we found that the nasolabial angle in skeletal Class III patients was more important for facial attractiveness than the relation of the upper lip to the E line (*Huang & Li, 2015*; *He, Gu & Sun, 2020*). This might be due to the commonly retracted position of the upper lip in skeletal Class III patients relative to the E-line. Consistent with previous research, we found that lower lip position and form significantly contribute to facial esthetics (*Hayashida et al., 2011*; *Sharma et al., 2022*).

With regard to subjective evaluation of profile changes during orthodontic treatment, we found that lower incisor position changes, including $\Delta$L1-AP, $\Delta$L1-NB, $\Delta$L1/NB, $\Delta$L1/AP and $\Delta$L1/MP, were significantly correlated with the profile change score. Moreover, changes in the lower lip-E line distance were more strongly correlated with the profile change score compared with changes in the upper lip, as represented by the nasolabial angle. An increase in overjet was beneficial to the profile VAS score, as most skeletal Class III patients have a negative overjet before treatment. However, it is also crucial not to increase the overjet *via* upper incisor proclination, as this could decrease the nasolabial angle and facial profile score.

Considering the challenges that camouflage orthodontic treatment presents for patients with prominent mandibular protrusion, we divided patients into two groups based on whether their mean SNB was >84°. The results revealed that for non-mandibular protrusion patients, retraction of both the upper and lower incisors was strongly correlated with the facial profile, as also observed in bimaxillary protrusion patients and those with Class I malocclusion (*Huang & Li, 2015*; *He, Gu & Sun, 2020*). However, in the mandibular protrusion group, only the lower incisor position change was significantly correlated with the facial profile. These results differed from those reported for skeletal Class II patients undergoing camouflage treatments, in which the change in U1-APo was significant to facial attractiveness (*Guo et al., 2023*). Previous studies have also stressed the importance of the distalization of the mandibular dentition for successful non-surgical treatment of skeletal Class III patients (*Lin & Gu, 2006*; *Yu et al., 2016*). Our findings underscore the importance of the lower incisor's lingual movement and change in the protrusive lower lip in achieving facial balance and harmony in mandibular protrusion patients.

We performed a stepwise multiple linear regression analysis to identify critical independent predictors of the posttreatment VAS scores of the facial profile in skeletal Class III patients. Pretreatment nasolabial angle, changes in the lower lip-E line distance, and pretreatment Pog-NB distance were the variables most strongly correlated with the posttreatment facial profile VAS score. Notably, among the independent predictors associated with orthodontic treatment, decreased distance between the lower lip and E line was strongly correlated with increased posttreatment facial profile score. A study of patients with bimaxillary protrusion reported that every millimeter of mandibular incisor retraction produced 0.6 mm of lower lip retraction (*Kusnoto & Kusnoto, 2001*). Similarly, total mandibular arch distalization led to the distal movement of the lower lip

in Class III malocclusion patients (*Yeon et al., 2022*). Therefore, our results indicate that several factors should be considered when providing camouflage orthodontic treatment for skeletal Class III patients. First, the pretreatment nasolabial angle should be assessed. A small pretreatment nasolabial angle may lead to a low posttreatment appearance score. In such cases, orthognathic surgery might be necessary (*Araujo & Squeff, 2021*). Next, the potential extent of lower lip retraction due to orthodontic treatment should be estimated, as the degree of lower lip soft tissue retraction is a crucial predictor of posttreatment facial esthetic score. Finally, inconsistency between skeletal parameters and soft-tissue analysis should be carefully considered when diagnosing skeletal Class III patients (*Nucera et al., 2017*). Skeletal pattern imbalance does not necessarily correspond to undesirable esthetics.

Potential growth might affect final results in adolescents, so we also compared differences between adolescent (initial age <18 years) and adult (initial age $\geq$ 18 years) patients. However, we observed no significant differences between these two groups. This might because the adolescents included in this study had passed the peak growth period (cervical vertebrae stages 4–6). Additionally, we did not apply orthopedic force during the camouflage treatment, so our treatment might not affect skeletal growth. Nevertheless, more work is needed to explore how patient growth affects camouflage treatments in Class III adolescent patients. We also found it challenging to group our study population to allow more detailed analyses of different types of skeletal Class III patients: we had a limited sample size, and numbers of males and females were not balanced. Therefore, larger multicenter studies will be needed to confirm our results.

## CONCLUSIONS

1. Prominence of the lower lip, nasolabial angle, and underlying lower incisor position are important variables in pretreatment facial profile evaluations of skeletal Class III patients undergoing camouflage orthodontic treatment.
2. Retraction of the lower incisor, a decrease in the Lower lip-E line distance and increase in the *Z* angle, interincisal angle, and overjet are important factors when evaluating facial profile changes in skeletal Class III patients receiving camouflage orthodontic treatment.
3. Pretreatment nasolabial angle, changes in the Lower lip-E line distance ($\Delta$Lower lip-E line), and pretreatment Pog-NB distance are crucial factors affecting posttreatment facial esthetics of skeletal Class III patients receiving camouflage orthodontic treatment.

### Funding

This study is financially supported by grants from the projects of the National Natural Science Foundations of China (No. 82170996) and Special Program for Series Clinical Research of Peking University School and Hospital of Stomatology (PKUSS-2023CRF302). The funders had no role in study design, data collection and analysis, decision to publish, or preparation of the manuscript.

## Grant Disclosures

The following grant information was disclosed by the authors:
National Natural Science Foundations of China: 82170996.
Special Program for Series Clinical Research of Peking University School.
Hospital of Stomatology: PKUSS-2023CRF302.

## Competing Interests

The authors declare there are no competing interests.

## Author Contributions

- Xiaobei Li conceived and designed the experiments, performed the experiments, analyzed the data, prepared figures and/or tables, authored or reviewed drafts of the article, and approved the final draft.
- Yuan Tian analyzed the data, prepared figures and/or tables, and approved the final draft.
- Runzhi Guo analyzed the data, prepared figures and/or tables, and approved the final draft.
- Weiran Li analyzed the data, authored or reviewed drafts of the article, and approved the final draft.
- Danqing He conceived and designed the experiments, authored or reviewed drafts of the article, and approved the final draft.
- Yannan Sun conceived and designed the experiments, authored or reviewed drafts of the article, and approved the final draft.

## Human Ethics

The following information was supplied relating to ethical approvals (i.e., approving body and any reference numbers):

The present study was approved by the Peking University School and Hospital of Stomatology ethics committee (PKUSSIRB-202168141), and all methods were performed in accordance with the Declaration of Helsinki. All patients provided verbal informed consent to participate in the study.

## Data Availability

The raw measurements are available in the Supplementary File.

## Supplemental Information

Supplemental information for this article can be found online at http://dx.doi.org/10.7717/peerj.17733#supplemental-information.

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
