# Peer review of "Evaluation of the facial profile of skeletal Class III patients undergoing camouflage orthodontic treatment: a retrospective study"

_PeerJ, doi:10.7717/peerj.17733_

## Round 0.1 · original submission · Major Revisions

All reviewers found the study interesting but raised concerns that will help you improve the study through revision. Please consider all reviewer's comments carefully. Additionally, please provide more details regarding the provided treatment (e.g. Did all patients receive treatment with fixed appliances in both jaws? or Did the patients wear any orthopedic appliances or skeletal anchorage aiming to change the skeletal configuration, or was the treatment focused primarily on dental changes?). Also, provide more information regarding the sample selection. Was there a consecutive search of the archives for patients fulfilling the eligibility criteria?

Reviewer 1 has requested that you cite specific references. You are welcome to add it/them if you believe they are relevant. However, you are not required to include these citations, and if you do not include them, this will not influence my decision.

Reviewer 1 ·

Basic reporting

.

Experimental design

.

Validity of the findings

.

Additional comments

I’ve extensively read the manuscript titled “Evaluation of facial profile in skeletal class III
patients undergoing camouflage orthodontic treatment”. The methodology is appropriate and quite linear with recent evidences/ studies on this topic. I’ ve not major concerns in this regard.

On the contrary, I’ve some concerns related to the implementation of the quality of the manuscript in terms of comparison with actual evidences. Below my humble considerations

A revision of scientific English language is required.
It is not clear which basic data the authors used as reference for sample size calculation. Please, provide more details
Did the researchers perform analysis of normality distribution of data findings? Same consideration for the equality of the variance
The discussion should be improved. I suggest the authors to better address other clinical implications of their data findings for example the inconsistency that can occur between skeletal parameters and soft-tissue analysis which, in turn, can influence the decision making process for both diagnosis and treatment planning in patients with sagittal discrepancies, especially when camouflage is an opportunity. Please, refer to appropriate evidence on this topic https://pubmed.ncbi.nlm.nih.gov/27932406/
Also, authors should better discuss about patients’ growth which in class III subjects can influence the treatment timing and the planning for orthodontic camouflage. This is another important aspect to consider.

Reviewer 2 ·

Basic reporting

This is an interesting study but some issues require further clarification or discussion.

Experimental design

The representation of men and women in the present sample was not balanced. Gender-based differences in the subjective evaluation of facial aesthetics are documented in the orthodontic literature.
Sample size was not calculated.
Were the photographs randomized before the assessment?
Were all cephalometric radiographs acquired with the same x-ray setup?

Validity of the findings

no comment

Additional comments

The limitations of the study should be discussed.
English language should be improved.

Reviewer 3 ·

Basic reporting

• The article is well-written. English language is professional and fluent.
• The literature is well referenced and relevant.
• Tables are relevant, well-designed and described.

Experimental design

• The research question could be better defined in the Introduction. Primary and secondary outcomes could be more clearly described.
• Age range seem to be very large, including adolescents and adults. This might be useful regarding age correlations to the studied parameters. Maybe the study of two different age groups, i.e., 13-17 years and 18-35 years could provide better insights, since adolescent have growth potentials that might affect the final results, depending on the treatment modality implemented. Also, descriptive statistics concerning age should be provided.
• Was camouflage achieved with fixed, or also with removable appliances that could affect skeletal growth, especially in adolescents?

Validity of the findings

• Was power analysis conducted?
• Were all data tested for the normality of the distribution? This should be mentioned in the manuscript. In case certain data did not follow the normal distribution, IQR and Spearman correlation coefficient should be implemented instead of SD and Pearson correlation coefficient.
• The data provided are informative and support the validity of the results.
• Thorough analysis and interpretation of the results was conducted.
• The conclusions are well-stated. The research question could be better clarified in the Introduction.

---

## Round 0.2 · Minor Revisions

The manuscript has been considerably improved through revision. Please consider few additional reviewer's comments carefully and perform thorough English language editing throughout the manuscript. Also, please provide more details regarding the implemented treatment (e.g. Did all patients receive treatment with fixed appliances in both jaws? or was the treatment focused primarily on dental changes?). Finally, provide more information regarding the sample selection. Was there a consecutive search of the archives for patients fulfilling the eligibility criteria?

**Language Note:** The Academic Editor has identified that the English language must be improved. PeerJ can provide language editing services - please contact us at [email protected] for pricing (be sure to provide your manuscript number and title). Alternatively, you should make your own arrangements to improve the language quality and provide details in your response letter. – PeerJ Staff

Reviewer 1 ·

Basic reporting

pertinent

Experimental design

pertinent

Validity of the findings

pertinent

Additional comments

the authors have successfully satisfied all my previous concerns. in my opinion the manuscript can be published

Reviewer 2 ·

Basic reporting

The manuscript has been considerably improved following the revision, but still some issues need to be resolved.
The research question could be better defined in the Introduction. The sentences “Several cephalometric measurements have been found to correlate …. after camouflage orthodontic treatment.” do not belong in this section.
Some editing for English language is still required, especially in the changed text after the revision.
Multiple in-text references to the same item should be separated with a semicolon.

Experimental design

Already reviewed.

Validity of the findings

Already reviewed.

---

## Round 0.3 · accepted · Accept

All reviewers' and editor's concerns have been adequately addressed during the revisions. The manuscript can be published in its current form.